# Impedance Measures for Detecting Electrical Responses during Acute Injury and Exposure of Compounds to Roots of Plants

**DOI:** 10.3390/mps5040056

**Published:** 2022-06-30

**Authors:** Robin Lewis Cooper, Matthew A. Thomas, David Nicholas McLetchie

**Affiliations:** Department of Biology, University of Kentucky, Lexington, KY 40506-0225, USA; matt137thomas@gmail.com (M.A.T.); mclet@uky.edu (D.N.M.)

**Keywords:** roots, electrophysiology, suprafusion, impedance, comparison of techniques

## Abstract

Electrical activity is widely used for assessing a plant’s response to an injury or environmental stimulus. Commonly, a differential electrode recording between silver wire leads with the reference wire connected to the soil, or a part of the plant, is used. One method uses KCl-filled glass electrodes placed into the plant, similar to recording membrane/cell potentials in animal tissues. This method is more susceptible to artifacts of equipment noise and photoelectric effects than an impedance measure. An impedance measure using stainless steel wires is not as susceptible to electrically induced noises. Impedance measurements are able to detect injury in plants as well as exposure of the roots to environmental compounds (glutamate). The impedance measures were performed in 5 different plants (tomato, eggplant, pepper, liverwort, and *Coleus scutellarioides*), and responses to mechanical movement of the plant, as well as injury, were recorded. Monitoring electrical activity in a plant that arises in a distant plant was also demonstrated using the impedance method. The purpose of this report is to illustrate the ease in using impedance measures for monitoring electrical signals from individual plants or aggregates of plants for potentially scaling for high throughput and monitoring controlled culturing and outdoor field environments.

## 1. Introduction

Measuring electrical responses in plants allows one to detect physiological changes taking place within a plant when movement of ions occurs. Electrical changes in plants occur with changes in hydration, response to chemical cues on roots or exposed tissue, photosynthetic processes or response to stressors such as physical injury or infection from foreign bodies, as well as a detection of a nutrient source [1,2,3,4]. The electrical signals can travel quickly to induce a local or a whole plant response [5,6].

The standard method to measure an electrical response in plants is to detect voltage changes with similar approaches used for detecting electrical changes across animal cells by intracellular recording techniques. The use of intracellular recording with glass electrodes and amplification is sensitive to field potentials and requires cumbersome equipment that limits portability. Impedance measures can detect electrical responses within a plant and are not as sensitive to field potentials in the environment. Impedance measures have been used in plants to detect changes in the environment which affects physiology [7]. Impedance measure in which the current is altered requires additional instrumentation to vary the current; however, an impedance converter measures the correlated impedance changes. An impedance converter provides a steady current between two leads and if there is a change in the resistance between the leads, due to changes in ionic movement within the plant, a change in the resistance is detected and relayed as an alteration in voltage, which can be detected [8].

Here, we compared measures of a standard intracellular glass microelectrode technique with an impedance measure implementing both surface and intracompartmental recordings, while highlighting advantages and disadvantages of each approach. Also illustrated are electrical responses measured with impedance due to plant injury and chemical cues in a variety of plant species.

The standard differential electrical measure detects a voltage change from a ground lead to a recording lead; however, if the recording lead does not have a high enough resistance, then small changes in the recording field are hard to detect. If both the ground and recording leads are immersed within a solution with a large surface exposure on the recording lead, then a low resistance input will result. Thus, having a small area of contact with the field being measured leads to a high resistance, and allows for small changes in current leading to a larger voltage change as established by Ohm’s law. A high resistance recording lead can be obtained by coating the lead with an insulator, while leaving a small amount of wire exposed at the tip, or placing the recording lead with more surface exposure within a glass microcapillary, which has a small tip opening to the media being measured. In this procedure, the microcapillary is filled with a conductive media, such as 3 M KCl or potassium acetate as typically used for recordings across cell membranes in animal tissue. For recordings within compartments and within cells of plants, a common practice is to use 0.3 or 0.1 M KCl within the recording glass microcapillary [1,2,9]. KCl is a good electrolyte for detecting voltage differences across compartments and across cell walls or membranes. This configuration for differential recording is susceptible to field potentials in the environment, such as 50 (Europe) or 60 (North America) Hz frequency from electronic equipment. Thus, a Faraday cage is commonly used for such recordings to shield the environmental electrical noise.

An impedance measure is similar to the differential recording mentioned above in that two leads are used to detect a change in the voltage. However, two leads are used to detect a change in resistance while passing a current. Impedance measures are used in various ways, including respiratory breathing rates with expansion and relaxation of a chest for mammals [10], the movements of a respiratory organ in crayfish to control aeration of gills [11], clinical neuromuscular diseases in mammals [12], heart rate of crustaceans submerged in water [13,14], as well as to detect when the environment causes physiological stress of crayfish, crab or shrimp [15]. Even the fine movements of a beating heart in larval *Drosophila* are able to be detected, as there is a wide range in the sensitivity with an impedance technique without being sensitive to surrounding electrical noise. Depending on how the measures are made, they can be noninvasive, such as a strap around the chest of a mammal, or two leads in the media to detect body movements of insect larvae [16,17]. With two leads in a media or solution with an organism or a tissue present, a small electrical field can be used. If there is any change in the resistance between the two leads, such as the movement of ions, a change can be detected. Herein, the use of impedance technique is implemented to measure electrical changes due to ionic movement within a plant during injury, and exposure to various compounds to the roots of the plant. Other measures using impedance can be recorded, such as the response of a healthy plant to stimuli and disease states, and ionic movements within the plant that occur during metabolic processes such as photosynthesis. Such measures are not only possible for acute changes within milliseconds, but monitoring long term recordings over days, weeks, and months are feasible.

Impedance measures have two advantages over the standard differential electrical measure. First, the burdensome electrical noise from equipment connected to wall outlets and room light fixtures when using differential recordings of electrical signals are not generally a problem when using impedance measures [13]. Second, the elimination of the photoelectric effect, due to the electrodes. Silver wires are normally used in differential electric measures due to the Ag-Cl junction which allows for a rapid exchange of electrons. Silver chloride leads are commonly used for monitoring current flow in the KCl solutions within glass microelectrodes. The photoelectric effect on silver wire due to light, and in particular white light, is well known in the field of electrophysiology, and always needs to be controlled when altering the lighting while recording electric potentials [18,19]. Impedance measures can use iridium:platinum wires or stainless steel wires, and these are not as susceptible to photoelectric effects as silver wires. Thus, one can avoid using silver leads for impedance measures to avoid a photoelectric effect. The use of an impedance converter does not require the more delicate and expensive equipment used in glass electrode intracellular recording methods.

The purpose of this report illustrates a non-invasive, low cost, portable, and relatively easy approach to monitor acute or chronic physiological changes in a plant resulting from ionic movements. In addition, a comparison of a few techniques in monitoring electrical signals within plants is presented. The favored approach presented is to place the impedance leads within the stem of a plant if the stem is large enough. For narrow stems, or when not wanting to pierce a plant, the use of a conductive paint to adhere the impedance leads to the plant stem is manageable. The impedance technique is less problematic than the use of glass electrodes with silver leads and standard intracellular amplifiers to monitor if a plant responds to a stimulus.

## 2. Procedure

### 2.1. Plants

Plants were obtained from a local supplier (Home Depot, Lexington, KY, USA) or raised in an environmentally controlled greenhouse at the University of Kentucky. Plants (tomatoes, banana peppers and eggplants) were all grown in potting soil from the supplier (Home Depot, Lexington, KY, USA). They were maintained in a laboratory in which the experiments were conducted for 2 weeks on a 12:12 light cycle with fluorescent lights. Plants (*Coleus scutellarioides*) from the greenhouse were grown in well trays in a Mycorrhizae potting soil mix (Premier HorticultureInc, Quebec Canada). We also used the liverwort *Marchantia inflexa* that was cultured in an environmentally controlled chamber (Percival, T-35LL) and consisted of a mat of thalli composed of multiple individual plants. A representative photo of each plant used is provided in Appendix B.

### 2.2. Recordings with a Differential Electrical Amplifier

Measuring the electrical response within the stems of the tomato plants was performed by inserting a glass microelectrode (catalogue # 30–31–0 from FHC, Brunswick, ME, USA, with the tip broken to a jagged opening in the range of 10 to 20 µM diameter) into the respective plant structure. The electrode was filled with 0.3 M KCl. A ground wire, used as a reference lead, was placed on the base of the stem. The surfaces of the plants were dry where the silver paint was adhered. The silver wire of the recording lead was inside a glass electrode, and the silver wire of the ground/reference leads were coated with chloride by using concentrated bleach for about 20 min to obtain the Ag-Cl coating. All wires were rinsed thoroughly with water prior to being used. The glass electrode was inserted into the stem using a micromanipulator under a dissecting microscope. The electrode was inserted 1 to 2 mm into the stem just across the first cell layer. The recordings were performed within a grounded Faraday cage. To reduce 60 Hz noise from the surroundings, a common ground wire with Ag-Cl coating was placed into the soil in which the plant was growing.

The electrical signals were obtained with an amplifier (Neuroprobe amplifier, A-M systems; obtained from ADInstruments, Colorado Springs, CO, USA) and connected to a computer via an AD converter (4s Power lab 4/26, ADInstruments, Colorado Springs, CO, USA). Recordings were performed at an acquisition rate of 20 kHz. Events were observed and analyzed with software Lab-Chart 8.0 (ADInstruments, Colorado Springs, CO, USA). 

### 2.3. Electrical Recordings with an Impedance Converter

The impedance technique was used for all plants (tomatoes, banana peppers and eggplants). Two insulated iridium-platinum wires (diameter 0.127 mm and with the coating 0.2032 mm; A-M Systems, Carlsburg, WA, USA) or insulated stainless steel wires (0.127 mm diameter and with coating 0.2032 mm diameter; A-M Systems, Carlsburg, WA, USA) were used. The iridium-platinum wires are more flexible than the stainless steel wires, but in penetrating the stem of the plant, the stainless steel wire was preferable due to its stiffness. In addition, the stainless steel wires were about a third of the cost. All data reported herein were with stainless steel wires. The insulation (~0.5 mm length) was removed with fire on the ends of both wires to be in contact with the plant. The other ends had the insulation removed (~1 cm) to be placed in the clamps of the impedance amplifier. The impedance amplifier (model 2991, UFI, Morro Bay, CA, USA) was used, which allowed changes in an electrical field to be monitored as a measure of dynamic resistance. The circuit diagram for the Model 2991 is provided on the company web site for the instrument (please see web page and the downloadable PDF from https://www.ufiservingscience.com/index.html (accessed on 20 May 2022); https://www.ufiservingscience.com/datasheets/2991manual.pdf (accessed on 20 May 2022)). The general arrangement of the instrument and a plant is as illustrated in Figure 1.

Two approaches were used for impedance measures. One approach involved placing the two leads along the stem of the plants with physical contact, but not penetrating the tissue. In this case, the conductive paint was applied sparingly over the exposed ends of the wires and on the plant. A second approach was to impale the stem of the plants with both leads to a depth of about 1 mm or less. For tomatoes, the two leads were 5 to 10 cm apart and for eggplant and peppers about 3 to 5 cm apart.

The output of the impedance amplifier was connected to a computer via an AD converter (4s Power lab 4/26, ADInstruments, Colorado Springs, CO, USA). Recordings were performed at an acquisition rate of 20 kHz for acute measures and at 100 points/sec for long term recordings over hours. Events were observed and recorded with software (Lab-Chart 8.0, ADInstruments, Colorado Springs, CO, USA).

### 2.4. Stimulus: Injury Induction and Environmental Changes

Because cutting the leaf results in leaf movement and tissue removal, we first bend the leaf to measure this stimulus, then we cut the leaf. The movement was performed by bending the leaf to the same degree as would occur by cutting the leaf. In some cases, leaves were taped to a supporting structure to avoid any movement of the stem where the recording leads were placed. Bending of a leaf generally provided some electrical responses which could be detected by all recording techniques used in this study. A cut in a leaf, by use of scissors, of each plant examined, was performed as a standard assay to determine if an injury response occurred which could be measured with electrophysiological recordings. The associated videos and figures illustrate some of the leaf bends and cuts performed.

Examining the ability of the impedance method to detect electrical changes in a plant due to surrounding environmental changes, glutamate (1 Molar) was dissolved in water and added to the soil which would bathe the roots of a tomato plant. A high concentration of glutamate is used as a proof of concept to examine if an electrical response was induced by the plant. The loose soil was approximately 200 mL in volume within a starter pot for gardens. 20 mL of solution was added to the top of the soil while recording from the plant for a 30 min period. Distilled water was used as a control for the effect of glutamate. In this paradigm the impedance wires were placed inside the stems.

## 3. Results

Tomato plants were used to examine the differences in recording techniques between the glass microelectrode method with associated intracellular amplifier, and the impedance method. Simultaneous recordings with these two techniques were made in the same plant while providing a given stimulus:bending and cutting a leaf (Figure 2A). An enlarged view shows one of the impedance leads with conductive paint (Figure 2B). Also shown is a glass microelectrode impaled into the stem with and the reference lead at the base of the stem held with conductive paint (Figure 1B).

A deflection of a leaf by bending produces an electrical signal detected by both the glass electrode and impedance leads as noted by the asterisk in Figure 3. A cut on a leaf also produced large defections in the electrical signals for both recording techniques (Figure 3). The processes of these manipulations along with computer traces are shown in video format (Appendix A; YouTube link; https://youtu.be/MA_fcOb_UMQ ( accessed on 20 May 2022)).

Thus, both recording techniques can monitor the electrical responses from the plant due to these stimuli. However, the use of the glass electrode approach and silver-chloride leads present artifactual noise when white light is used, which is in part due to the photoelectric effect (Figure 4). With the light, not only is 60 Hz noise present but masked by the noise is the electrical offset due to the photoelectric effect on the silver leads (Figure 4A1). If the trace is filtered by a running average of 501 points in the 20 kHz acquired trace, the 60 Hz is smoothed out and the offset due to the photoelectric effect is obvious (Figure 3B1). Note the electrical trace for the impedance measure is not altered by the 60 Hz induced by the light fixture or by a photoelectric effect (Figure 4A2,B2).

One issue with impaling the plant with a glass electrode is that initial electrical potential will gradually show a depolarization for a few to several minutes before becoming stable. This is likely due to the injury and possibly the initial and slow leak of 0.3 M KCl from the solution used inside the glass electrode to contact the silver wire for transferring the electrical signal. The surface recording with the impedance did not generate a gradual depolarization due to the resistance remaining the same over time, and no injury was induced in the plant.

The same plant was used with the leads of the impedance converter attached to the surface of tomato stem with conductive paint and then subsequently inserting the two impedance leads within the stem about 1 mm. For both techniques, the leaf was bent a few times prior to making a cut on the leaf (Figure 5A,B). The eggplant model was used for this demonstrative purpose due to the ease of placing the impedance electrodes into the soft stem of the eggplant.

Leaves which were to be cut were first subjected to leaf-bending controls which mimicked the amount of movement that would occur during a leaf cut. These controls were then compared to the leaf wound responses and the wounding responses were found to be larger and more rapid. To examine the effect on the electrical signal of bending a leaf to about the same extent as the movement caused when making a cut on a leaf, leaves were examined on a plant to be tested for the injury of a cut. Then the same leaf as performed for the bending was also used to make a cut to induce an injury discharge. An example of this procedure is shown in video format with an eggplant and a pepper plant (Appendix A; Youtube; https://youtu.be/a-2nWWA3iNU (access on 20 May 2022)). This procedure was performed in six different plants for *Coleus scutellarioides* (Figure 6), tomatoes (Figure 7), eggplant (Figure 8) and peppers (Figure 9). In each recording, a cut produced a larger and more rapid response (*p* < 0.05, *N* = 6 for each species, rank sum Wilcoxon test). Note that each plant produced different amplitude responses for bending a leaf and for a cut. Since each plant was different in morphology and size it is expected that there would be variation observed among plants and species. The rationale for showing the six trails for each species is to illustrate that relative changes for each plant might be more practical for comparison between plants for the same stimulus. The asterisk shown in each recording demarcates a bending of the leaf as a test for the effect of movement on the leaf which occurs while cutting a leaf.

When performing the cuts to the various plants, larger defections were observed to be associated with the cutting of the main vein within the leaves compared to cutting minor veins within a leaf. As an illustration, an eggplant was used, and a cut was made in a leaf where only a minor vein was cut. In using the same plant but a leaf on the opposite side at about the same distance from the leads was used to cut across the main vein on the leaf (Figure 10).

To illustrate the usefulness of long term recordings using the impedance method, a tomato plant was recorded continuously for 6 h with the impedance lead impaled into the plant. The acquisition rate was at 100 points per second. Thus, rapid changes of a few milliseconds would not be detected. A leaf cut of the same degree but on different leaves at the same height on the plant was performed initially and again after the 6 h of recording for comparisons in responses. (Figure 11). It was interesting to note that over the 6 h recording period some large defections in the recordings were present as well as some oscillations in the wave form (not shown). These oscillations were not analyzed for this study as it is beyond the scope of this methodological report. Such signals would be of interest for future studies.

In examining if the impedance measures within the stem above the soil were sufficient to detect changes in the environment in which roots of a plant are exposed, a flow through the soil experiment was performed for the effect of water and glutamate as a proof of concept. The roots of many species of plants are sensitive to glutamate [20]. The soil was moist when starting this protocol but not saturated. Upon adding water to the surface of the soil an electrical response was detected followed by a burst of signals two minutes afterwards (Figure 12). The water was able to be flushed through the soil and leave the container the plant was grown in. After 10 min, the same volume (100 mL) of solution was poured over the soil but this time containing L-glutamate (1 M). This resulted in a rapid burst of activity which diminished over time. After 10 min, water (100 mL) was added in the same manner. This resulted in an even larger electrical response. With a second flush of water (100 mL) after 10 min, the response was still robust. Only after a third flushing with water (100 mL) did the response start to dampen. With the fourth flush of water (100 mL) it is likely most of the glutamate was washed through the soil and off the roots.

In the circumstance in which aggregates of individual plants or a colony of a plant (i.e., Aspen trees),when the roots or rhizoids are interwoven and contained in the same media, it is potentially possible to measure electrical responses by the impedance approach to examine the effects of stimuli, such as an injury or disease within the aggregate. To illustrate this possibility, a mat of liverworts (*Marchantia inflexa*) were utilized. The two impedance leads were placed 2 cm apart into thallus tissue, and a cut was made to a third thallus between the two leads (Figure 13).

## 4. Discussion

The methods presented illustrate commonly used approaches to record electrical events in plants. The intracellular recording technique commonly used for animal cells with glass microcapillaries and a KCl solution to a silver lead in order to compare to a reference lead was shown to be very sensitive to electrical signals in the plants but also sensitive to the surrounding environment, and generally requires a Faraday cage to block the surrounding field potentials. In addition, the silver-chloride wires are susceptible to the photoelectric effect by a change in light. The two approaches with impedance measures with the electrode lead either adhered to the plant with a conductive paint, or placing the leads directly into the plant, indicated that placing the leads within the stem of the plants provided a better signal to noise response. The impedance technique provides a relatively easy approach with insulated stainless steel wires to record electrical activity in a plant due to chemical cues in the roots, mechanical movements of the leaves as well as an injury to a leaf of the plant. The ease in recording impedance measures without glass microelectrodes, micromanipulators and intracellular amplifiers is cost effective (~800 USD for an impedance converter; model 2991, UFI, Morro Bay, CA, USA). Considering a microelectrode puller is needed to make glass electrodes, secondary equipment is required with this approach, whereas the impedance method requires no secondary equipment. This makes the impedance method suitable for students, amateur enthusiasts, and those on a budget who may lack specialized equipment needed for the glass microelectrode method. Furthermore, the impedance measures can be obtained without being influenced by electrical surrounding noise or photoelectric effects making it a suitable technique to record electrical signals within plants without the need for Faraday cages or light curtains. Finally, this recording technique requires simple hardware (i.e., AD board) making the impedance method ideal for measuring plant responses in the field with a data logger which can later be downloaded to a computer. Such instrumentation is now available commercially from various suppliers.

Impaling leads into the plant possibly obtains a better signal/noise ratio compared to surface leads due to the direct contact with the fluids within the stem of the plants as opposed to the contact through and between the cells on the surface of the stem to the internal fluids. Plants with large cells, such as green alga *Chara*, have made it easier in the past to know exactly where one is obtaining a recording (i.e., intracellular or extracellular compartments) [21]. Surface electrodes at least provide some overview in the field potential as well as a lead placed within the plant but not in a specific location. If plants with bark or thick dry outer layers were to be used, one could potentially make a small hole for leads to be placed inside the tree in an area in contact with the phloem and possibly even the xylem. These holes are likely necessary for recording tree responses as leads on the surface with conductive paint may not be in electrical contact with the fluids within the plant. The strength of the electrical signals within the plant would also make a difference in the need to obtain the most refined approach necessary. New techniques are being developed to have electrical contact with plants for recording potentials which may allow surface recordings to have a higher conductivity with a plant [22]. It would be of use to examine the ionic make-up in dehydrated as well as hydrated plants to determine if the signals would vary depending on the ionic strength of the media within the plant. Examining signals when plants become dormant or non-dormant can be easily studied.

The variation in responses within a plant species to approximately the same stimulus of either bending of a leaf or cutting a leaf illustrates the complexity in quantifying the responses to a given stimulus. A leaf bend in a species gave very different amplitude signals as well as a leaf cut, but within a plant the cut produces larger and rapid responses compared to the response from bending. The types of ion channels activated by bending a leaf as compared to bending while cutting may be a rationale for inducing local differences, in addition to altering the resistance within the plant with a cutting of the fluid filled veins on the leaf. It would be of interest to know the types and density of stretch activated channels on a leaf, types of leaves and differences among species of plants [23,24,25]. The measurement responds to a change in resistance with the impedance converter instrument, providing a voltage difference between the two wires. Such a difference varies from plant to plant as the amount of resistance varies based on the plant’s hydration and likely which compartment the leads are placed within the vascular plants. This is difficult to control for in laboratory experiments as well as in the field. The point made in this study is that one can measure changes to note an injury, such as a cut in a leaf or stem, and if the part of the plant is being bent. The degree of change is different for each plant and among plants. However, the advantage of a glass electrode is that a direct measure in the electrical potential differences can be obtained, but this will also vary from plant to plant if the ground is placed in or on the plant, in the soil around the plant and depending on the distance from the plant. The moisture of the soil or on the surface of the plant as well as the ionic make of the media would also result in variability in the measures among plants. The stimuli used in this study (i.e., a cut of a leaf, bending of a leaf) produce different responses for each plant being measured. The cuts provided a larger response than bending of a leaf in each case, but the degree of change varied among plants of the same species and of the different species. Thus, the measures are relative for each plant and species and each environment within the plant.

Perhaps within a plant or a recording arrangement a known stimulus is needed to compare changes to detect other types of stimuli [26]. This is likely possible with computational software, larger sample sizes, and automated artificial intelligence analysis. In considering making electrical recordings in the field, both the microelectrode and impedance techniques detect leaf movements with just a single leaf being bent out of the many on the plant, so to determine an injury induced response over movements by wind or contact by another plant would still need to be investigated if the resolution of an injury or response to a chemical cue over background signals can be determined. Cuts to primary, secondary, and tertiary veins producing different magnitude responses can be further addressed in potential plants with compound leaves. If one leaf is connected to the stem closer to the recording electrode, the signal may be larger than for leaves farther away from the recording leads.

The ability to record for prolonged periods of time would allow various types of experimental manipulations to be measured, which could take time to be manifested by the plant. Since exposure to compounds such as the amino acid glutamate produced a measurable response, plants can be screened in a relatively high throughput for sensitivity to various compounds which generate electrical responses. Measures within a group of plants in an aggregate, or within an aquaculture arrangement, may be possible to assess a group effect to determine if one plant within a group is injured while monitoring a surrounding field by many plants. Because electrical responses vary for injured or diseased plants [2,26], potentially signals with particular signatures can be correlated to particular plants [9,26,27]. This would depend on the intensity of the signal and the conductivity of the media the plants are in.

There are many future experiments of interest still to be addressed in the field of plant electrophysiology, such as if electrical signals are responsible for the induction of hormones and other chemical signals within and among plants, and what type of molecular changes may occur due to such electrical signals. Once one understands the characteristics of electrical signals in specific plants to various cues, it would be of interest to induce such electrical signatures into plants and determine how plants respond to artificially induced signals as compared to the naturally induced ones.

As a proof of concept, this report has demonstrated that an impedance measure with an impedance converter is feasible to detect electrical signals with good resolution and with limited interfering electronic noise or photoelectric effects. The resolution of the recordings does depend on the gain used in the impedance measures. As more current between the leads is applied, the ability to detect changes in resistance will vary. Each recording system used would need to be adjusted for detecting the stimuli of interest. However, the impedance technique is not as useful to obtain absolute potentials across cell membranes or walls as a differential electrical measure. Thus, depending on what measure is needed, this report has provided some examples in the use of impedance measures for various experimental needs.

## Figures and Tables

**Figure 1 mps-05-00056-f001:**
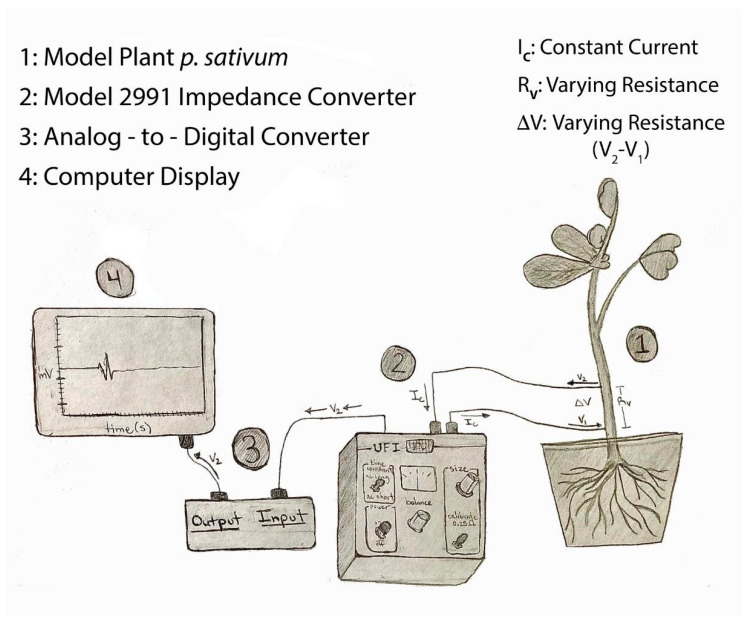
A schematic of the recording arrangement for impedance measurements with a plant being monitored for a change in resistance.

**Figure 2 mps-05-00056-f002:**
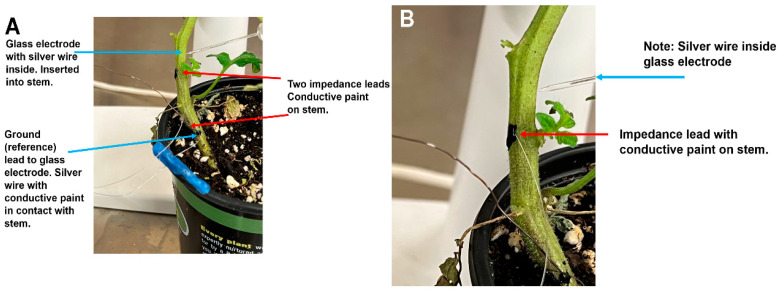
(**A**) Illustration of the simultaneous recording approach of an intracellular glass electrode and reference lead as well as an approach used for a surface recording along the stem of a tomato plant with impedance. (**B**) An enlarged view of the glass microelectrode placed inside the stem with the silver wire inside the glass electrode and one of the impedance leads attached to the surface of the stem with conductive paint.

**Figure 3 mps-05-00056-f003:**
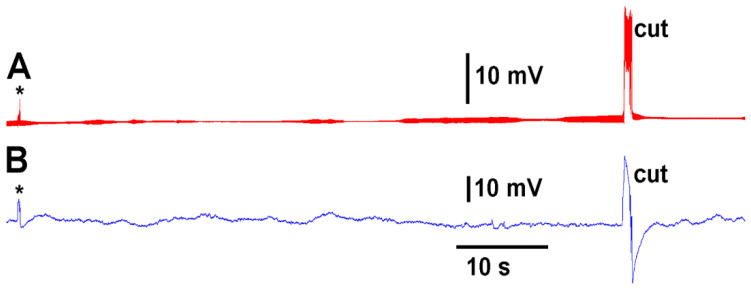
The two approaches in recording electrical responses simultaneously in a tomato plant. (**A**) The electrical signal obtained with a glass microelectrode in the stem of the plant. (**B**) The electrical signals obtained with surface leads and use of an impedance converter. The asterisk demarcates a bending of the leaf as a test for the effect of movement on the leaf. The cut made on the tip of the leaf was readily observed in both recording approaches.

**Figure 4 mps-05-00056-f004:**
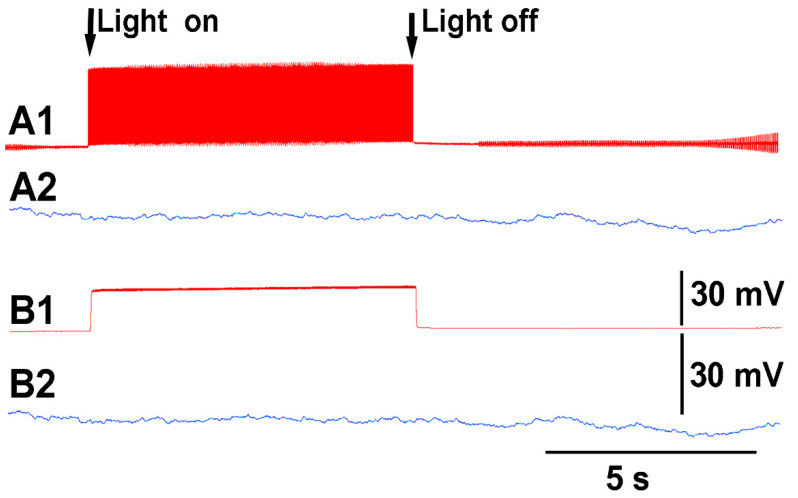
The induced noise from electronic equipment and photoelectric effect in altering the signals obtained with glass microelectrode and silver wire technique as compared to impedance measures in simultaneous recording within a given tomato plant. (**A1**) Signals obtained with the glass microelectrode and silver wire within the 0.3 M KCl electrode solution. (**B1**) The same signal with taking a running average of every 501 data points. (**A2**,**B2**) The impedance measure made with surface electrodes attached to the plant with conductive paint. The averaged trace in B1 illustrates the photoelectric effect induced by light on the silver leads used for the glass microelectrode approach.

**Figure 5 mps-05-00056-f005:**
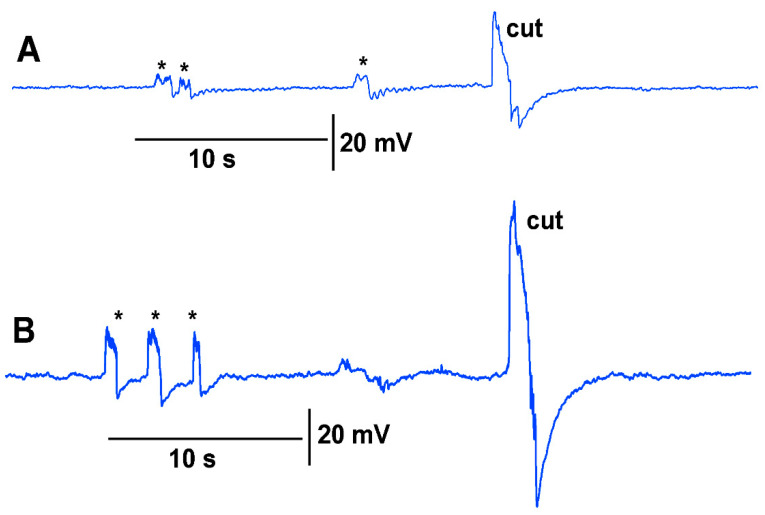
Comparison of impedance using surface contact or tissue penetration of the stem while bending and cutting leaves. Penetration of the plant with the leads shows a more robust signal. (**A**) The two leads of the impedance converter were adhered to the surface of the stem with conductive paint while moving the leaf and cutting a leaf. (**B**) The same plant used for the recording in A was used and the leads were placed within the stem about 1 mm next to the same locations as used for the surface measurement. The asterisk (*) demarcates a bending of the leaf as a test for the effect of movement on the leaf which occurs while cutting a leaf.

**Figure 6 mps-05-00056-f006:**
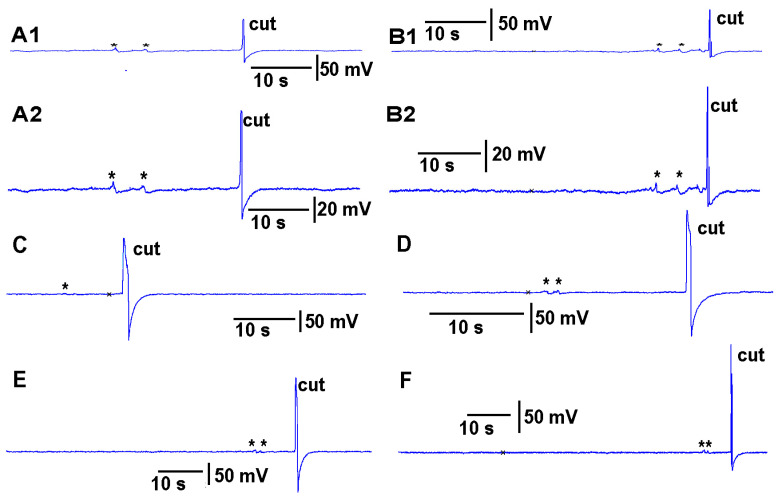
The effect of bending a leaf and cutting the same leaf while measuring the impedance changes by recording with both leads placed within the stem of the plant *Coleus scutellarioides*. The asterisk shown in each recording demarcates a bending of the leaf as a test for the effect of movement on the leaf which occurs while cutting a leaf. (**A1**,**B1**–**F**) illustrates six different plants. (**A1**,**B1**) are presented at the same scale as (**C**–**F**) for comparisons, whereas (**A2**,**B2**) are enlarged to better note the changes in the responses.

**Figure 7 mps-05-00056-f007:**
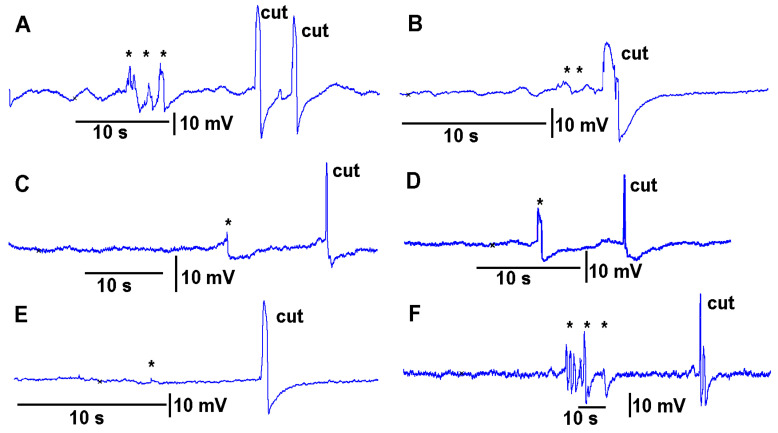
The effect of bending a leaf and cutting the same leaf while measuring the impedance changes by recording with both leads placed within the stem of tomato plants. (**A**–**F**) illustrates six different plants. The asterisk (*) shown in each recording demarcates a bending of the leaf as a test for the effect of movement on the leaf which occurs while cutting a leaf.

**Figure 8 mps-05-00056-f008:**
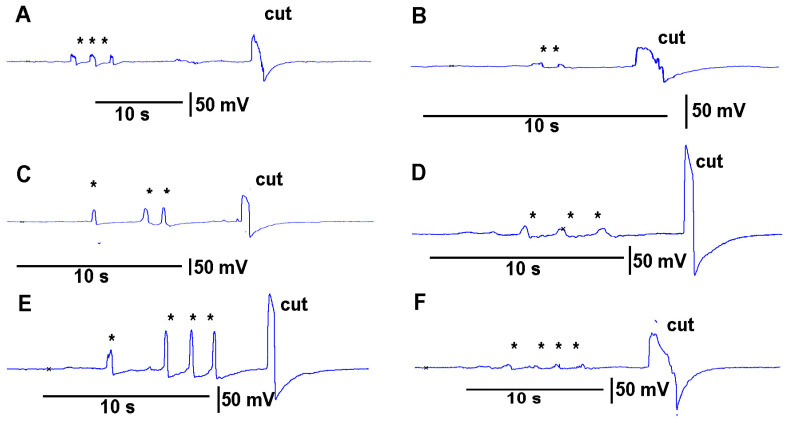
The effect of bending a leaf and cutting the same leaf while measuring the impedance changes by recording with both leads placed within the stem of eggplants. (**A**–**F**) illustrates six different plants.The asterisk (*) shown in each recording demarcates a bending of the leaf as a test for the effect of movement on the leaf which occurs while cutting a leaf.

**Figure 9 mps-05-00056-f009:**
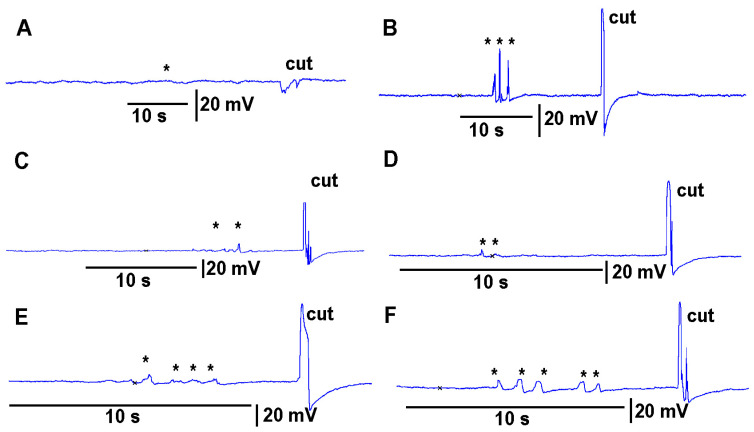
The effect of bending a leaf and cutting the same leaf while measuring the impedance changes by recording with both leads placed within the stem of pepper plants. (**A**–**F**) illustrates six different plants. The asterisk (*) shown in each recording demarcates a bending of the leaf as a test for the effect of movement on the leaf which occurs while cutting a leaf.

**Figure 10 mps-05-00056-f010:**
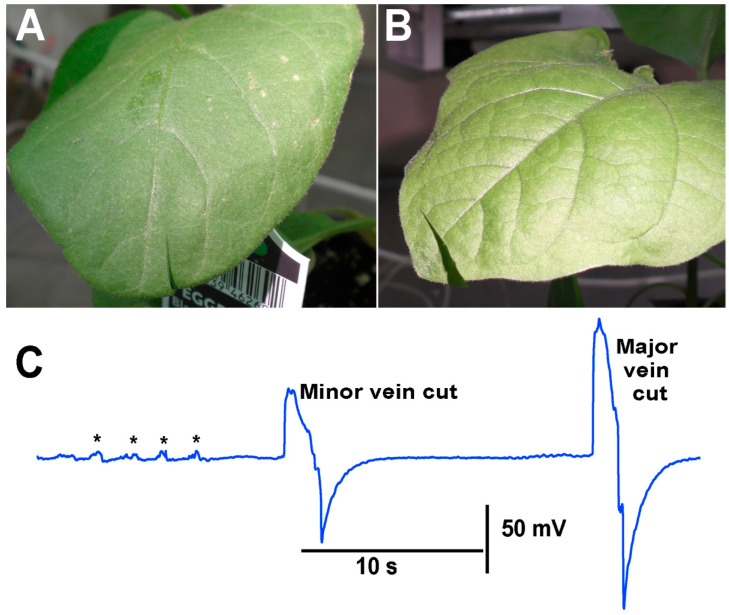
Electrical responses when cutting minor veins as compared to a major vein within a leaf of an eggplant. (**A**) A cut made across a minor vein on a leaf. (**B**) In a leaf on the opposite side of the plant from the leaf shown in A a leaf was cut across the major vein. (**C**) The electrical signal obtained with the impedance measure with leads placed within the stem of an eggplant while making leaf cuts. The first cut was from the leaf shown in A and the second cut was made from the leaf shown in B. The asterisk (*) shown in each recording demarcates a bending of the leaf.

**Figure 11 mps-05-00056-f011:**
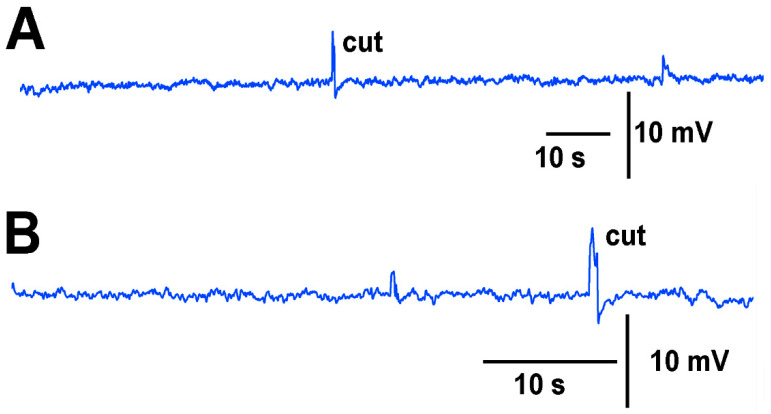
Long term recordings performed with impedance measures in a tomato plant. The leads of the wires were placed within the stem of a tomato plant. (**A**) After 5 min of recording a leaf tip was cut. (**B**) After 6 h of continuous recording another leaf tip was cut of the same degree, at about the same height on the plant.

**Figure 12 mps-05-00056-f012:**
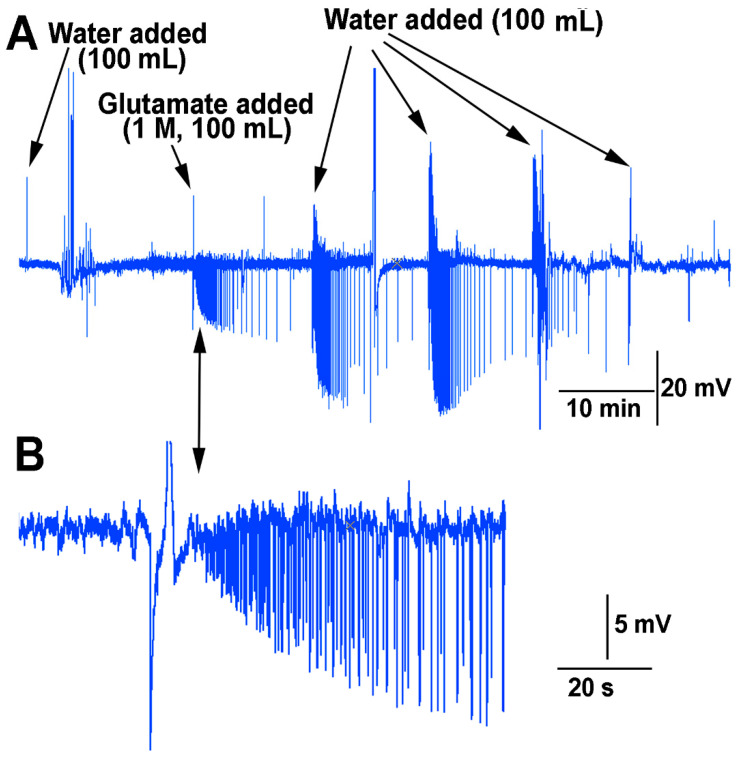
Impedance measures while exposing the roots of a tomato plant to water and L-glutamate (1 M). (**A**) The application of water to damp soil produced electrical responses. The electrical responses were greatly enhanced by the exposure of the roots to L-glutamate (1 M). Flushing the glutamate through the soil with subsequent additions of water still produced large changes in amplitudes and frequency of responses until the glutamate was finally flushed through the soil. (**B**) An enlarged view of the first response to exposure of the roots to glutamate. This illustrates the initial high frequency of signals which decreases in time.

**Figure 13 mps-05-00056-f013:**
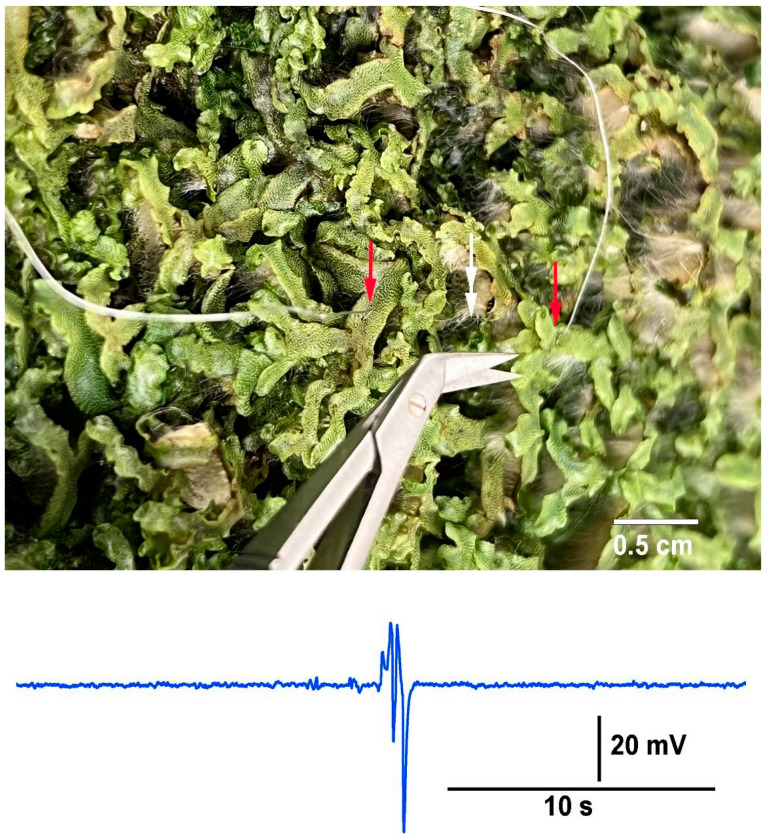
Electrical response to an injury of a neighboring plant in an aggregate of liverworts *Marchantia inflexa*. The electrical signal was detected by the impedance leads which were impaled into the thallus of two plants 5 cm apart from each other. The red single arrow heads indicate where the leads were placed, and the white double arrowhead indicates where the cut was made to one thallus halfway in between the two leads.

## Data Availability

Not applicable.

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
