# Peer review of "Impedance Measures for Detecting Electrical Responses during Acute Injury and Exposure of Compounds to Roots of Plants"

_mps, 2022, doi:10.3390/mps5040056_

Round 1

Reviewer 1 Report

This work presents an alternative method for monitoring the plant activity, at cell level, based on the use of the bioimpedance as biomarker. It is proposed a two-wires setup for impedance test on plants, as alternative to classic patch-clamp methods, that measure the ionic activity inside the cells. The methods it is applied to several plants, obtaning a very interesting response to injuries provoked, demostrating it feasibility.

Novelty: Main contribution of the paper are the experimemental results obtained. It is relevant, for a electronics engineering as me, to see this kind of electric answer from plants. The assays have been well designed and described, and the proposed method is welll validated.

Main comment: from the proposed setup point of view, the full electronic system should be better described at Material and Methods section, may be introducing a new figure for electronic system description:

- Detailled image of electrodes employed. 

- Electrical system connection diagram and operation.

In this sense, the two-wires system employed required always calibration if exact bioimpedance under test must be known. Could the authors describe in more detail how the test system perform each measure?

- What are really the impedance values measure? In the paper are described the plants responses as voltages-versus-time only.. but the plant biomarker is associated to each bioimpedance tested.

Minor conments:

- Please, explain the sentence:

"An impedance converter will provide a DC voltage due to impedance changes from changes in resistance, capacitance, or inductance variations, as well as a combination of these which is ready easy to record and amplify"

A DC voltage is ussually considered as a voltage constant in time.

- Upper case in unit (KHz) is reserved for Kelvin. Change to kHz.

- In Figures 5 and 7, the voltages scales should be the same for the six plant, to better perception of the plant to plant response changes.

- It possible to see the 6h assays in figure 10?

In my opinion, the paper must be rewriten in the section relative to the electronic system description before to be considered to be accepted.

Author Response

We greatly appreciate the comments and suggestions by the reviewer.

We agree with the reviewer  with the suggested changes.

We hope that each point is addressed satisfactorily as we want this study to be a useful study for others in future investigations.

  1. This work presents an alternative method for monitoring the plant activity, at cell level, based on the use of the bioimpedance as biomarker. It is proposed a two-wires setup for impedance test on plants, as alternative to classic patch-clamp methods, that measure the ionic activity inside the cells. The methods it is applied to several plants, obtaning a very interesting response to injuries provoked, demostrating it feasibility.

Response to reviewer: Thank you. We hope other researchers may find a benefit in this report.

  1. Novelty: Main contribution of the paper are the experimemental results obtained. It is relevant, for a electronics engineering as me, to see this kind of electric answer from plants. The assays have been well designed and described, and the proposed method is welll validated.

Response to reviewer: Thank you. We welcome input and suggestions from a person as yourself with knowledge of electronics.

Main comment: from the proposed setup point of view, the full electronic system should be better described at Material and Methods section, may be introducing a new figure for electronic system description:

- Detailled image of electrodes employed. 

Response to reviewer: This is a good point. However, the instrument used is from a company and we did not design the instrument. The full circuit description is provided on the company web site for the instrument we used the Model 2991 (please see  web page and PDF)

https://www.ufiservingscience.com/index.html

So, we will state the source for the wiring diagram and more information about the instrument.

- Electrical system connection diagram and operation.

Response to reviewer: We refer the reader to the user manual and referenced it now in the revised version. We did include an new figure as a schematic of  the recording arrangement for the impedance measure.

In this sense, the two-wires system employed required always calibration if exact bioimpedance under test must be known. Could the authors describe in more detail how the test system perform each measure?

Response to reviewer: Ok, good suggestion if it was not clear. We revised and described this point in more detail. The measured responses in a change of resistance with the impedance converter instrument provides a voltage difference between the two wires. Such a difference varies from plant to plant as the amount of resistance varies on the plant being hydrated or not and likely which compartment the leads are placed within the vascular plants. This is difficult to control for in laboratory experiments as well as in the field. The point made in this study is that one can measure changes to note an injury such as a cut in a leaf or stem and if the a part of the plant is being bent. The degree of change is different for each plant and among plants. However, the advantages of a glass electrode is a direct measure in the electrical potential differences can be obtained, but this will also vary from plant to plant if the ground is placed in or on the plant, in the soil around the plant and distance from the plant. The moisture of the soil or on the surface of the plant as well as the ionic make of the media would also result in variability in the measures among plants. Thus, the measures are relative for each plant and each environment. This is explained now in the revised text within the Discussion.

- What are really the impedance values measure? In the paper are described the plants responses as voltages-versus-time only.. but the plant biomarker is associated to each bioimpedance tested.

Response to reviewer: Ok, good suggestion. We are not sure what the reviewer means by the biomarker in this situation. If the reference is to the degree of injury (i.e., the leaf cut) or the bending of a leaf this depends again on the size of the plant and relation to hydration and ionic makeup of the plant fluid the leads are measuring. Thus, the measures are relative for each plant and each environment. This is explained now in the revised text within the Discussion. We revised and described this point in more detail

Minor conments:

- Please, explain the sentence:

"An impedance converter will provide a DC voltage due to impedance changes from changes in resistance, capacitance, or inductance variations, as well as a combination of these which is ready easy to record and amplify"

Response to reviewer: Ok. We used the manufacture’s description in the original text. We have modified to exampling this in a more straight forward manner.

“An impedance converter provides a steady current between two leads and if there is a change in the resistance between the leads, due to changes in ionic movement, a change in the resistance is detected and relayed as a voltage change to be detected [8]. “

A DC voltage is ussually considered as a voltage constant in time.

Response to reviewer: Yes we agree. These are represented on the scale bars for the figures.

- Upper case in unit (KHz) is reserved for Kelvin. Change to kHz.

Response to reviewer: Good catch. Correction made. Thank you

- In Figures 5 and 7, the voltages scales should be the same for the six plant, to better perception of the plant to plant response changes.

Response to reviewer: These figures are now corrected to be at the same scaling. Good idea.

We also show a higher resolution with the preparations with smaller responses. Figure 8 was also changed to maintain a similar Y-Axis scale

- It possible to see the 6h assays in figure 10?

Response to reviewer: We are not sure what is meant here. Is it that you want the traces shown over the 6 hours ?  Part B is after 6 Hours from the same recording. If the request to show all 8 hours, we can show multiple lines like over each 30 min  and 12 lines, but this would be a figure that would take a whole page and the responses would be hard to see in such a compressed figure.  Or do you want a movie recording of the assay over 6 hours or a movie with the 1st part (bend/cut) and then a short movie after 6 hours (bend/cut)?

In my opinion, the paper must be rewriten in the section relative to the electronic system description before to be considered to be accepted.

Response to reviewer: Ok. But the electronics of the instruments is all on the manufactures web page of the instrument. We would just be reproducing their circuit diagram (we might need permission to copy their figures/description).  Maybe best just to direct the reader to the company web site with the full details.

https://www.ufiservingscience.com/index.html

Reviewer 2 Report

1. The authors should thoroughly revise the language of the manuscript. Many language and format problems can be found in the manuscript. 

2. In the section 2, Procedure, the authors suggest to include some illustrations/pictures to help clarify the methods. 

3. What's the major improvements of the proposed methods compared to the conventional techniques? The authors suggest to quantify them in the manuscript. 

Author Response

We greatly appreciate the comments and suggestions by the reviewer.

We agree with the reviewer  with the suggested changes.

We hope that each point is addressed satisfactorily as we want this study to be a useful study for others in future investigations.

The authors should thoroughly revise the language of the manuscript. Many language and format problems can be found in the manuscript. 

Response to reviewer: We did revise the text and found some typos. We also provided better descriptions of the equipment and measures made. We welcome any points where the reviewer feels the text is not well presented.

In the section 2, Procedure, the authors suggest to include some illustrations/pictures to help clarify the methods. 

Response to reviewer: Thank you for the suggestions. We do have detailed set up description in Figure 1 and two YOUTUBE movies to show how the measures are made and results obtained. It was possible the YouTube links were not hot linked in the 1st submission. The YouTube links should show the details as requested. We took some of the figures shown in the YouTube to be placed in the manuscript for ease in understanding the methods.

What's the major improvements of the proposed methods compared to the conventional techniques? The authors suggest to quantify them in the manuscript. 

Response to reviewer: Each experiment conducted they were repeated at least 6 times, using a different plant each time. The electrical responses are presented in the figures to scale in this revised version. The variation is now discussed in the Discussion section.

Round 2

Reviewer 1 Report

The questions have been positively answered.

Reviewer 2 Report

The authors have addressed all the comments. I do not have further comments.